# ADAPTIVE LEARNING OF QUANTUM HAMILTONIANS

## ABSTRACT

The challenge of learning representations for quantum Hamiltonian systems resides at the intersection of quantum information and learning theory. Viewed through the lens of learning theory, this task can be regarded as the non-commutative counterpart to learning graphical models. In our research, we design and analyze adaptive learning algorithms, including the quantum iterative scaling algorithm (QIS) and gradient descent (GD), for the Hamiltonian inference problem using adaptive Gibbs state oracles. Our principal technical contribution centers on a thorough analysis of their convergence rates, involving the establishment of both lower and upper bounds on the spectral radius of the Jacobian matrix for each iteration of these algorithms. Furthermore, we explore quasi-Newton methods to enhance the performance of both QIS and GD. Specifically, we propose using Anderson mixing and the L-BFGS method for QIS and GD, respectively. These quasi-Newton techniques exhibit remarkable efficiency gains, resulting in orders of magnitude improvements in performance.

## 1 INTRODUCTION

The Hamiltonian of a quantum system holds a central role in quantum mechanics, governing the system's time evolution as dictated by Schrödinger's equation as well as the system's Gibbs state at thermal equilibrium. Consequently, the task of learning the Hamiltonian for a specific quantum system represents an important research topic in areas including quantum information, quantum machine learning, and condensed matter physics. Algorithms designed for this learning task are instrumental in characterizing and verifying quantum many-body systems and quantum computing devices emerging with the active development of quantum hardware (Arute et al., 2019; Wu et al., 2021).

In this paper, we consider adaptive learning algorithms designed for what we refer to as the Hamiltonian inference problem, a variant of the Hamiltonian learning problem extensively studied in the literature (Wiebe et al., 2014a;b; Evans et al., 2019; Wang et al., 2017; Anshu et al., 2021; Amin et al., 2018; Bairey et al., 2019; 2020; Haah et al., 2022). In the Hamiltonian learning problem discussed in Anshu et al. (2021), for example, the objective is to determine the Hamiltonian description, denoted as $H = \sum_j \mu_j H_j$, by consuming copies of the Gibbs states $\xi_H$ associated with the corresponding Hamiltonian.[1] In contrast, the Hamiltonian inference problem involves the computation of the Hamiltonian from the input $\alpha_j = \langle H_j \otimes \mathbb{1}, \xi_H \rangle$ using *adaptive* access to a Gibbs state preparation oracle for Hamiltonians chosen by the learning algorithm from a linear family of Hamiltonians having the same geometric connectivity as $H$. Two key distinctions set these two problems apart. Firstly, the Hamiltonian inference problem requires a more powerful oracle capable of generating the Gibbs state of a Hamiltonian chosen by the algorithm. In contrast, Hamiltonian learning only uses the Gibbs states for the true Hamiltonian. Secondly, unlike the Hamiltonian learning problem, the Hamiltonian inference problem does not assume access to the entire Gibbs state of the true Hamiltonian. For example, it does not have information about quantities like $\langle O, \xi_H \rangle$ for operators not in the span of $H_j$. This information may become crucial for certain approaches to Hamiltonian learning (Qi & Ranard, 2019; Bairey et al., 2019; 2020). These two distinctions render the two problems closely related but non-comparable.

In classical machine learning contexts, a related problem pertains to graphical models, specifically concerning the learning of Markov random fields (Kindermann & Snell, 1980) or Boltzmann ma-

---

[1]The Gibbs state $\xi_H$ is $\frac{1}{Z}e^{-\beta H}$ where $Z$ is the partition function $\operatorname{tr} e^{-\beta H}$ normalizing the state.

chines (Ackley et al., 1985). The Hammersley-Clifford theorem establishes that any positive probability distribution satisfying the local Markov property can be represented as a Gibbs distribution (Lafferty et al., 2001). Hence, learning the model is equivalent to learning the parametrization of a classical Hamiltonian defining the Gibbs distribution (Li & Korb, 2020). In recent years, sample-efficient and time-efficient algorithms with demonstrated effectiveness for specific classical graphical models have emerged (Ravikumar et al., 2010; Bresler, 2015; Hamilton et al., 2017; Klivans & Meka, 2017; Vuffray et al., 2016). However, the challenge of extending these classical techniques into the quantum domain remains non-trivial, with ongoing nascent efforts in this direction (Anshu et al., 2021; Haah et al., 2022).

Both the Hamiltonian inference problem and Hamiltonian learning problem are closely related to the maximum entropy problem through the well-known Jaynes' principle in statistical physics (Jaynes, 1957). Consider a family of Hamiltonian operators $H(\mu) = \sum_{j=1}^{m} \mu_j H_j$, parameterized by a vector of real numbers $\mu = (\mu_j)_{j=1}^{m}$ where the local terms $H_j$ satisfy the condition $\|H_j\| \leq 1$. Given as input the average values $\alpha = (\alpha_j)_{j=1}^{m}$, Jaynes' principle ensures that $\alpha$ uniquely determines the parameter vector $\mu$ in the sense that $\langle H_j, \xi_{H(\mu)} \rangle = \alpha_j$ has unique solutions for $\mu$ where $\xi_{H(\mu)} = \frac{1}{Z(\mu)} e^{-\beta H(\mu)}$ is the Gibbs state for Hamiltonian $H(\mu)$. Furthermore, the Gibbs state $\xi_{H(\mu)}$ is the solution to the maximum entropy problem below in Fig. 1. In this sense, the Hamiltonian inference problem is essentially the *computational version* of the Jaynes' principle that aims to compute $\mu$ given $\alpha$ as input.

| Maximum entropy problem | Dual problem |
|---|---|
| maximize: $S(\rho)$ | minimize: $\ln \operatorname{tr} \exp\left(\sum_j \lambda_j H_j\right) - \lambda \cdot \alpha$ |
| subject to: $\langle H_j, \rho \rangle = \alpha_j$, | |
| $\rho$ is a density matrix, | subject to: $\lambda_j \in \mathbb{R}$. |

Figure 1: The maximum entropy problem and its dual problem. Here, $S(\rho) = -\operatorname{tr}(\rho \ln \rho)$ is the von Neumann entropy of $\rho$.

Previous works have mainly focused on the Hamiltonian learning problem. Building upon Jaynes' principle and several other important techniques, it was established in a recent breakthrough (Anshu et al., 2021) that the *sample complexity* of the Hamiltonian learning problem exhibits polynomial scaling with the number of qubits for quantum systems with geometric locality and a constant temperature. The proof of this result relied on proving the strong convexity of the log-partition function $\ln Z(\mu)$, as expressed by the inequality $\nabla_\mu^2 \ln Z(\mu) \succeq \gamma I$ where $\gamma$ is a positive number. This result will also be utilized to establish the convergence rate for algorithms considered in our paper for the Hamiltonian inference problem. In another line of research, the Hamiltonian learning problem was reduced to a linear equation problem (Bairey et al., 2019; 2020). The method is expected to work under certain conditions of the Hamiltonian, yet a rigorous description of the condition and complexity analysis for the algorithm to work are still unknown.

Given that the Hamiltonian inference problem is essentially the maximum entropy problem (Fig. 1), it is a natural attempt to generalize classical learning algorithms for maximum entropy problems to the quantum setting. Classically, generalized iterative scaling (GIS) and a close variant known as Improved Iterative Scaling (IIS) are the most studied algorithms (Darroch & Ratcliff, 1972; Della Pietra et al., 1997). These algorithms arise from statistics research and have rigorous analysis in terms of their convergence proofs. For practical applications, however, quasi-Newton methods are known to be the state-of-the-art solutions to the maximum entropy inference problem and parameter learning of graphical models (Malouf, 2002). In this work, we explore quantum generalizations of both the GIS algorithm and quasi-Newton methods for solving the Hamiltonian inference problem.

## 1.1 OUR CONTRIBUTIONS

This paper offers three key contributions. Firstly, we present the convergence rate analysis for both Quantum Iterative Scaling (QIS) and Gradient Descent (GD) algorithms. This is achieved by representing the iteration's Jacobian in a concise and explicit formula, which extends a fundamental result from Liang et al. (2004). Secondly, we establish bounds on the eigenvalues of the Jacobian, proving

polynomial convergence of both the QIS and GD algorithms. Lastly, we investigate two variants of quasi-Newton methods that accelerate the iterative processes for QIS and GD, respectively, resulting in significant performance improvements. In the following, we delve into a detailed discussion of these three contributions.

### 1.1.1 EXPLICIT JACOBIAN FORMULA FOR ITERATIONS

The QIS algorithm we consider here is introduced in Ji (2022), and serves as a natural quantum counterpart to the GIS algorithm. It is presented as a specialized variant of the approximate matrix Legendre-Bregman projection algorithm, and the convergence proof of QIS was established by using the auxiliary function method and matrix inequalities (Ji, 2022). However, while the auxiliary function method is versatile, it falls short in providing a precise assessment of the algorithms' convergence rates. To solve this problem, we perform a more tailored analysis of the QIS algorithm's convergence rate in this paper. In the classical case, a corresponding convergence analysis for the GIS algorithm is detailed in Liang et al. (2004). This classical analysis leverages Ostrowski's theorem, which bounds the convergence rate of an iterative procedure by the spectral radius of the Jacobian matrix associated with the iteration. Our contribution extends the classical analysis to the quantum setting and addresses the difficulty that the gradient of the exponential function for matrices is much more involved than the classical counterpart. We provide a closed-form formula for the Jacobian of the QIS iteration, generalizing the approach of Liang et al. (2004). The formula has a concise form $\mathbb{1} - P^{-1}L$ where $P$ is a diagonal matrix whose entries are the mean values of operators related to $H_j$ over the Gibbs state and $L$ is the Hessian of the log-partition function. This connection to the Hessian matrix $L$ is crucial for the convergence proof.

The maximum entropy problem has a dual problem, which is an unconstrained optimization problem regarding the log-partition function in Fig. 1. As a comparison, we consider the gradient descent (GD) algorithm for the dual problem. The Jacobian of the GD update process can be computed as $\mathbb{1} - \eta L$ where $L$ is the Hessian of the log-partition function and $\eta$ is the step size of GD. In a sense, the Jacobian of the QIS update rule, $\mathbb{1} - P^{-1}L$, can be seen as a mechanism to *adaptively* choose the step size for different directions. This is the main advantage that QIS has over the GD algorithm. Numerical simulations in Fig. 2 of Section 6 also show that QIS converges significantly faster than GD.

### 1.1.2 UPPER AND LOWER BOUNDS FOR THE JACOBIAN

As our main technical contribution, we analyze the eigenvalues of the Jacobian matrix by establishing both the lower and upper bounds for them.

First, we prove that all eigenvalues of the Jacobian are non-negative. This result is established by proving an *upper bound* on the Hessian of the log-partition function $L \preceq P$. The main difficulty for proving such a bound arises from the fact that there is no simple explicit formula for the derivative of the matrix exponential function $\frac{\mathrm{d}}{\mathrm{d}s}e^{H+sV}$. Hasting's quantum belief propagation (Hastings, 2007) expresses the derivative $\frac{\mathrm{d}}{\mathrm{d}s}e^{H+sV}$ as the anti-commutator $\{e^{H+sV}, \Phi(V)\}$ for some quantum channel $\Phi$ depending on $H + sV$ and is the main technical tool used in many previous works for addressing this difficulty. However, this form of quantum belief propagation is not applicable in our case to prove the inequality because the anti-commutator form only guarantees the Hermitian property of the derivative, while the inequality requires positivity. We propose a modified quantum belief propagation operator (see Lemma C.3) to circumvent the problem.

Second, we show that the largest eigenvalue of the Jacobian is bounded away from 1 by making a connection to the strong positivity of the log-partition function proven in Anshu et al. (2021), a *lower bound* on the Hessian of the log-partition function. The upper and lower bounds of the Jacobian together complete the convergence rate analysis of the QIS algorithm by using Ostrowski's theorem (Theorem B.1).

### 1.1.3 ACCELERATIONS

While the QIS and GD algorithms enjoy provable convergence analysis and are expected to converge in a polynomial number of iterations for local Hamiltonians at constant temperature, it can exhibit sluggish performance in practical scenarios. For classical learning of model parameters,

quasi-Newton methods are recommended for solving the maximum entropy inference problems, as suggested in a systematic comparison of classical maximum entropy inference algorithms performed in Malouf (2002). Even though the convergence analysis is usually less established, quasi-Newton methods are usually much faster in practice than iterative scaling and gradient descent algorithms. In light of this, we investigate two families of quasi-Newton accelerations of the algorithms.

The first family of heuristic acceleration is based on the Anderson mixing method (Anderson, 1965). The Anderson mixing algorithm is a heuristic method for accelerating slow fixed-point iterative algorithms. It can be seamlessly integrated to work with the QIS algorithm as QIS is indeed a fixed-point iteration. The Anderson mixing accelerated QIS algorithm (AM-QIS) has *exactly the same* computational requirement of the QIS algorithm in terms of the oracle access and the type of measurements required on the quantum system. The second family is based on the BFGS method (Nocedal & Wright, 2006; Yuan, 2015) and in particular the limited memory variant, L-BFGS is applied to the GD algorithm (L-BFGS-GD). In our numerical simulations, we observed that AM-QIS and L-BFGS-GD have comparable performance, usually faster by orders of magnitude than the standard QIS and GD algorithms.

We believe that applying such quasi-Newton heuristics is important for quantum optimization algorithms like the QIS algorithm considered here. While quantum computing offers a promising new paradigm with the potential for substantial speedups in specific problems, the practical construction of large-scale quantum computers is still in its early stages. Current quantum computing technology has limitations in terms of scale and suffers from errors. Hence, quantum computing power remains a scarce and valuable resource. Given this scenario, the careful optimization of resources required to solve problems on quantum computers emerges as a critical task. The use of Anderson mixing and BFGS for Hamiltonian inference algorithms and potentially for other fixed-point iterative quantum algorithms represents an attempt to achieve such resource optimization. Notably, this approach is not unique to quantum computing. In fact, the quasi-Newton method, which developed into an important optimization heuristics, was initially developed by W. Davidon while working with early classical computers, which often crashed before producing correct results. In response, he devised faster heuristics to expedite calculations later known as the first quasi-Newton method! Quantum computers are currently in its very early stages. They are unstable and prone to errors just like classical computers in the early days; hence, such heuristic speedups may be critical for numerical quantum algorithms.

## 2 PRELIMINARY

In this section, we introduce some notations used in this paper. For two real vectors $x, y \in \mathbb{R}^m$, we define $x \cdot y$ as $\sum_{i=1}^m x_i y_i$. We sometimes extend this notation to the case when $y$ is a vector of matrices and write, for example, $\lambda \cdot F$ to mean the summation $\sum_j \lambda_j F_j$. For matrices $A, B$, define $\langle A, B \rangle = \text{tr}(A^\dagger B)$. We use $A \succeq B$ or $B \preceq A$ to mean $A - B$ is a positive semidefinite matrix. A density matrix $\rho$ is a positive semidefinite matrix of unit trace. The set of density matrices on Hilbert space $\mathcal{X}$ is denoted $\text{D}(\mathcal{X})$.

Suppose $\mathcal{X}$ is a finite-dimensional Hilbert space and $f$ is a real convex function. We use $\Delta$ to denote the domain $\text{dom} f$ of $f$, the interval on which $f$ takes well-defined finite values. Then $f$ extends to all Hermitian operators in $\text{Herm}_\Delta(\mathcal{X})$ as $f(X) = \sum_k f(\lambda_k)\Pi_k$ where $X = \sum_k \lambda_k \Pi_k$ is the spectral decomposition of $X$. Denote the interior and boundary of $\Delta$ as $\Delta_{\text{int}}$ and $\Delta_{\text{bd}} = \Delta \setminus \Delta_{\text{int}}$ respectively. It is easy to see that the domain of matrix function $f$ is $\text{Herm}_\Delta(\mathcal{X})$, and the interior of the domain is $\text{Herm}_{\Delta_{\text{int}}}(\mathcal{X})$.

Given convex function $f$ as above, the Bregman divergence for matrices is $D_f(X, Y) = \text{tr}(f(X) - f(Y) - f'(Y)(X - Y))$, where $X \in \text{Herm}_\Delta(\mathcal{X})$ and $Y \in \text{Herm}_{\Delta_{\text{int}}}(\mathcal{X})$. An important case we focus on in this paper is $f(x) = x \ln x - x$. In this case, the matrix Bregman divergence becomes the Kullback-Leibler divergence $D(X, Y) = \text{tr}(X \ln X - X \ln Y - X + Y)$ defined for non-normalized matrices $X, Y$. When $X, Y$ are positive semidefinite matrices of trace 1, it recovers the Kullback-Leibler relative entropy $D(X, Y) = \text{tr}(X \ln X - X \ln Y)$. We will need the matrix Bregman-Legendre projection $\mathcal{L}(Y, \Lambda)$ and Bregman-Legendre conjugate $\ell(Y, \Lambda)$ for convex function $f(x) = x \ln x - x$ defined as $\mathcal{L}(Y, \Lambda) = \exp(\ln Y + \Lambda)$, $\ell(Y, \Lambda) = \text{tr} \exp(\ln Y + \Lambda) - \text{tr} Y$. For $Y \propto \mathbb{1}$, $\ell(Y, \Lambda) = \text{tr} \exp(\Lambda)$ and we omit $Y$ and write it as $\ell(\Lambda)$.

In this paper, we consider spin Hamiltonians only and write them as $H = \sum_{j=1}^{m} H_j$ where $H_j$'s are local terms acting on at most constant number of neighboring qubits according to certain interaction geometry. For example, a $ZZ$ term acting on the first two qubits is $Z \otimes Z \otimes (\mathbb{1}^{\otimes n-2})$ for Pauli operator $Z = \begin{pmatrix} 1 & 0 \\ 0 & -1 \end{pmatrix}$. We often use $\xi_H = \frac{1}{Z} e^{-\beta H}$ to represent the Gibbs state of the Hamiltonian $H$ for inverse temperature $\beta$ specified in the context or $\beta = 1$ otherwise. Here $Z = \operatorname{tr} e^{-\beta H}$ is the partition function normalizing the state to have trace 1 and plays an important role in statistical physics and also in our work.

## 3 QUANTUM ITERATIVE SCALING

This section presents a version of the Quantum Iterative Scaling (QIS) algorithm introduced in Ji (2022) and discusses its applications in the Hamiltonian inference problem.

We first introduce some notations used in the following discussions. For a given list of Hermitian matrices $F = (F_j)_{j=1}^{m}$, define the linear family of quantum states $\mathscr{L}(\rho_0)$ as $\{\rho \succeq 0 \mid \langle F_j, \rho \rangle = \langle F_j, \rho_0 \rangle\}$. Define the exponential family $\mathscr{E}(\sigma_0)$ as $\{\frac{1}{Z} \exp(\ln \sigma_0 + \lambda \cdot F)\}$. We introduce the new notation $F_j$ playing the role of $H_j$ in the previous discussion as we will need certain normalization conditions. In the end, we will choose $F_j = \frac{H_j + \mathbb{1}}{2m}$ so $F_j$ is a scaled linear shift of $H_j$ such that $F_j \succeq 0$ and $\sum_j F_j \preceq \mathbb{1}$.

We note that, in Ji (2022), the algorithms are designed for non-normalized matrices and, therefore, there is no need to explicitly normalize $Y^{(t)}$ in the update. Here, we perform explicit normalization to work with normalized quantum states and their von Neumann entropy. For $Y^{(t)} = \exp(\ln \sigma_0 + \lambda \cdot F)$, the normalization is equivalent to a linear update in the summation of the exponential function $Y^{(t)} = \exp(\ln \sigma_0 + \lambda \cdot F - \ln Z)$ where $Z = \operatorname{tr} Y^{(t)}$. Hence, we have the following two methods to handle the normalization. The first is to let the algorithm to find the normalization implicitly, and this would require that $\mathbb{1}$ is in the span of the $F_j$'s. This will be the case if the assumption on $F_j$ is that $\sum_j F_j = \mathbb{1}$. The second is to perform the normalization explicitly as we did in the algorithm. This approach is advantageous as it works for all $F_j$'s satisfying $\sum_j F_j \preceq \mathbb{1}$ even if $\mathbb{1}$ is not in the span of $F_j$'s.

An important special case of the algorithm is when $\sigma_0 = \mathbb{1}/d$ and $D(\rho, \sigma_0) = \ln(d) - S(\rho)$ where $d$ is the dimension. Then, the minimization over the linear family is now exactly the maximum entropy problem as in Fig. 1 with $H_j = F_j$, $\alpha_j = \langle F_j, \rho_0 \rangle$, and $\lambda_j = -\beta \mu_j$. When all the operators $F_j$'s are diagonal in the computational basis, the QIS algorithm recovers the GIS algorithm (see e.g. Theorem 5.2 of Csiszár & Shields (2004)).

---

**Algorithm 1** Quantum iterative scaling algorithm.

---

**Require:** $\rho_0, \sigma_0 \in \mathrm{D}(\mathcal{X})$ such that $D(\rho_0, \sigma_0) < \infty$.
**Input:** $F = (F_1, F_2, \ldots, F_k) \in \mathrm{Pos}(\mathcal{X})^k$ and $\sum_{j=1}^{k} F_j \preceq \mathbb{1}$.
**Output:** $\lambda^{(1)}, \lambda^{(2)}, \cdots$ such that

$$\lim_{t \to \infty} D(\rho_0, \mathcal{L}(\sigma_0, \lambda^{(t)} \cdot F)) = \inf_{\lambda \in \mathbb{R}^k} D(\rho_0, \mathcal{L}(\sigma_0, \lambda \cdot F)).$$

1: Initialize $\lambda^{(1)} = (0, 0, \ldots, 0)$.
2: **for** $t = 1, 2, \ldots,$ **do**
3:     Compute $Y^{(t)} = \mathcal{L}(\sigma_0, \lambda^{(t)} \cdot F)$.
4:     **for** $j = 1, 2, \ldots, k$ **do**
5:         $\delta_j^{(t)} = \ln \langle F_j, \rho_0 \rangle - \ln \langle F_j, Y^{(t)}/\operatorname{tr} Y^{(t)} \rangle$.
6:     **end for**
7:     Update parameters $\lambda^{(t+1)} = \lambda^{(t)} + \delta^{(t)}$.
8: **end for**

---

To give some intuition behind the QIS algorithm, we define $\xi^{(t)} = Y^{(t)}/\operatorname{tr} Y^{(t)}$ and note that the update in the QIS algorithm is simply $\delta_j = \ln \langle F_j, \rho_0 \rangle - \ln \langle F_j, \xi^{(t)} \rangle$, which is zero when the

linear family constraint $\langle F_j, \rho \rangle = \langle F_j, \rho_0 \rangle$ is satisfied by $\rho = \xi^{(t)}$. In this case, the algorithm stops updating $\lambda$ in the $j$-th direction as expected. Otherwise, if the difference between the current mean value $\langle F_j, \xi^{(t)} \rangle$ and the target value $\langle F_j, \rho_0 \rangle$ is big, so will be the update $\delta_j$. The algorithm is, in this sense, *adaptive* when compared to algorithms like multiplicative weight update algorithms (Arora et al., 2012).

The maximum entropy problem has dual program in Fig. 1 which is an unconstrained problem. Hence, it is also attractive to work with the dual using the gradient descent method (or corresponding quasi-Newton methods discussed later in the paper). The gradient of the dual objective function is $\frac{\partial}{\partial \lambda_j} \big( \ln \ell(\lambda \cdot F) - \lambda \cdot \alpha \big) = \langle F_j, \xi_{\lambda \cdot F} \rangle - \alpha_j$, where $\xi_{\lambda \cdot F}$ is the Gibbs state for Hamiltonian $\lambda \cdot F$. Therefore, in gradient descent, the update in each step is $\eta \big( \alpha_j - \langle F_j, \xi_{\lambda \cdot F} \rangle \big)$, where $\eta$ is the learning rate. This leads to the gradient descent algorithm in Algorithm 2.

---

**Algorithm 2** Gradient descent algorithm for Kullback-Leibler divergence minimization.

**Require:** $\rho_0, \sigma_0 \in D(\mathcal{X})$ such that $D(\rho_0, \sigma_0) < \infty$.
**Input:** $F = (F_1, F_2, \ldots, F_k) \in \mathrm{Pos}(\mathcal{X})^k$ and $\sum_{j=1}^{k} F_j \preceq \mathbb{1}$.
**Output:** $\lambda^{(1)}, \lambda^{(2)}, \cdots$ such that

$$\lim_{t \to \infty} D\big(\rho_0, \mathcal{L}(\sigma_0, \lambda^{(t)} \cdot F)\big) = \inf_{\lambda \in \mathbb{R}^k} D\big(\rho_0, \mathcal{L}(\sigma_0, \lambda \cdot F)\big).$$

1: Initialize $\lambda^{(1)} = (0, 0, \ldots, 0)$.
2: **for** $t = 1, 2, \ldots,$ **do**
3:     Compute $Y^{(t)} = \mathcal{L}(\sigma_0, \lambda^{(t)} \cdot F)$.
4:     **for** $j = 1, 2, \ldots, k$ **do**
5:         $\delta_j^{(t)} = \eta \langle F_j, \rho_0 \rangle - \eta \langle F_j, Y^{(t)} / \mathrm{tr}\, Y^{(t)} \rangle$.
6:     **end for**
7:     Update parameters $\lambda^{(t+1)} = \lambda^{(t)} + \delta^{(t)}$.
8: **end for**

---

The QIS algorithm generally outperforms the bare-bones GD algorithm. Consider the update of the QIS algorithm $\ln \alpha_j - \ln \langle F_j, \xi_{\lambda \cdot F} \rangle$, and the update of the GD algorithm $\eta \big( \alpha_j - \langle F_j, \xi_{\lambda \cdot F} \rangle \big)$, For $\alpha_j$ and $\langle F_j, \xi_{\lambda \cdot F} \rangle$ in $(0, 1]$, the QIS update is more aggressive than the dual gradient descent for learning rate $\eta \leq 1$ while still guarantees the convergence. This effect is more evident when the two numbers $\alpha_j$ and $\langle F_j, \xi_{\lambda \cdot F} \rangle$ are small, which holds in most applications. We will later see that choosing an appropriate learning rate will improve the performance of the GD algorithm considered in Section 4, but it is still less efficient compared to QIS.

## 4 CONVERGENCE RATE

In this section, we analyze the geometric convergence rate for the QIS algorithm in Algorithm 1 for the case when $\sigma_0 = \mathbb{1}/d$. As a comparison, we will also analyze the convergence rate of the GD algorithm (Algorithm 2).

We will come across several matrices which are defined here for later references. For $\lambda$ and $F$, we define the corresponding Gibbs state as $\xi = \dfrac{\exp(\lambda \cdot F)}{\mathrm{tr} \exp(\lambda \cdot F)}$, and for an operator $O$, we use $\langle O \rangle = \mathrm{tr}(O\xi)$ to mean the average value of $O$ with respect to $\xi$. Define diagonal matrix $P = \sum_j \langle F_j \rangle |j\rangle\langle j|$. Finally, define $L$ to be the Hessian of the log-partition function $\ln \mathrm{tr} \exp(\lambda \cdot F)$ with $\lambda$ as the variables.

We have the following two results regarding the QIS and GD algorithms respectively, proved in Appendix B.

**Theorem 4.1.** *The Jacobian of the iterative update map $\lambda^{(t)} \mapsto \lambda^{(t+1)}$ of Algorithm 1 for $\sigma_0 = \mathbb{1}/d$ is given by $\mathbb{1} - P^{-1}L$ for $P$ and $L$ defined above with $\lambda = \lambda^{(t)}$.*

**Theorem 4.2.** *The Jacobian of the iterative update map $\lambda^{(t)} \mapsto \lambda^{(t+1)}$ of Algorithm 2 is given by $\mathbb{1} - \eta L$ where $L$ is the Hessian of the log-partition function defined above.*

By the Ostrowski theorem stated in Theorem B.1, the geometric convergence rate of the QIS algorithm is, therefore, governed by the spectral radius of the Jacobian $\mathbb{1} - P^{-1}L$. Hence, we need to prove bounds on the spectral radius. In Anshu et al. (2021), a non-trivial lower bound on $L$ is proved (see Theorem C.2), giving an upper bound of the spectral radius. To prove the lower bound, will need the following result proved in Appendix C.

**Theorem 4.3.** *Let $P$ and $L$ be matrices defined above. We have $L \preceq P$.*

We are now able to state the convergence rate of the QIS and GD algorithms.

**Theorem 4.4.** *For Hamiltonian $H = \sum_{j=1}^{m} \mu_j H_j$ where conditions of Theorem C.2 are satisfied and $H_j$ are traceless terms with norm $\|H_j\| \leq 1$, the QIS algorithm with $F_j = \frac{\mathbb{1}+H_j}{2m}$ and the GD algorithm with the same choices of $F_j$ and $\eta = m$ solve the Hamiltonian inference problem with geometric convergence rate $1 - \Omega\left(\frac{1}{m^2}\right)$.*

*Proof of Theorem 4.4.* We consider the QIS algorithm first, which is the more difficult case. By Theorem 4.1, $J_{\text{QIS}} = \mathbb{1} - P^{-1}L$. Hence, we can bound the spectral radius as

$$
\begin{aligned}
r(J_{\text{QIS}}) &= r(\mathbb{1} - P^{-1}L) \\
&= r(\mathbb{1} - L^{1/2}P^{-1}L^{1/2}) \\
&= \left\|\mathbb{1} - L^{1/2}P^{-1}L^{1/2}\right\|
\end{aligned}
$$

Here, the second line follows from the fact that $AB$ and $BA$ have the same set of eigenvalues and the third step follows as $L^{1/2}P^{-1}L^{1/2}$ is Hermitian. By definition, $P$ is a diagonal matrix whose $(j,j)$-th entry is $\langle F_j, \xi \rangle = \left\langle \frac{\mathbb{1}+H_j}{2m}, \xi \right\rangle \leq 1/m$. Theorem 4.3 then implies that $\mathbb{1} - P^{-1}L$ has eigenvalues in $[0,1]$ and

$$
r(J_{\text{QIS}}) = 1 - \lambda_{\min}(L^{1/2}P^{-1}L^{1/2}) \leq 1 - m\lambda_{\min}(L). \tag{1}
$$

The Hamiltonian is $H(\mu) = \sum_j \mu_j(2mF_j - \mathbb{1})$. Define $\lambda_j = 2m\mu_j$, and $\tilde{H}(\lambda) = \sum_j \lambda_j F_j$. We have $\tilde{H}(\lambda) = H(\mu) + \mu_\Sigma \mathbb{1}$ where $\mu_\Sigma = \sum_j \mu_j = \sum_j \lambda_j/(2m)$. We compute

$$
\begin{aligned}
L_{j,k} &= \frac{\partial^2 \ln \text{tr} \exp(\lambda \cdot F)}{\partial \lambda_j \, \partial \lambda_k} \\
&= \frac{\partial^2 \ln \text{tr} \exp(\lambda \cdot F)}{\partial \mu_j \, \partial \mu_k} \frac{\partial \mu_j}{\partial \lambda_j} \frac{\partial \mu_k}{\partial \lambda_k} \\
&= \frac{1}{4m^2} \frac{\partial^2 \left( \ln \text{tr} \exp(H(\mu)) + \sum_j \mu_j \right)}{\partial \mu_j \, \partial \mu_k} \\
&= \frac{1}{4m^2} \nabla_\mu^2 \ln \text{tr} \exp(H(\mu)).
\end{aligned}
$$

By Theorem C.2, we have $\lambda_{\min}(L) \geq \Omega\left(\frac{1}{m^3}\right)$. Together with Eq. (1), this completes the proof using Theorem B.1.

By a similar calculation, we can prove the claim for the GD algorithm. $\qquad\square$

The above analysis shows that the QIS algorithm has a better geometric convergence rate even if we set $\eta = m$ in the GD algorithm. Numerical simulations in Section 6 also confirm this observation. In some sense, the QIS algorithm is an adaptive gradient descent that can automatically choose the appropriate learning rate for different dimensions as $\langle F_j \rangle$ may differ for each $j$.

## 5 ACCELERATION BY QUASI-NEWTON METHODS

The convergence analysis in Section 4 is of theoretical interest but polynomial convergence proved there is usually not enough for practical applications. In this section, we explore the application of quasi-Newton methods, which can significantly improve the efficiency of the adaptive learning

algorithms considered in Section 3. In particular, we study two families of methods, the Anderson mixing (Anderson, 1965) method and the BFGS method (Nocedal & Wright, 2006).

Anderson mixing (abbreviated as AM in the following) is a widely used method employed in numerical and computational mathematics to accelerate the convergence of fixed-point iterations. It particularly excels in scenarios where traditional iterative methods may converge slowly or struggle to find solutions efficiently. The essence of the Anderson mixing algorithm lies in its ability to dynamically combine and update a finite set of historical iterates. It adaptively selects a linear combination of these historical iterates, leveraging the past information to guide the algorithm toward convergence more effectively. This technique finds applications in various scientific and engineering domains, including quantum chemistry (Garza & Scuseria, 2012), machine learning (Sun et al., 2021), and solving complex systems of equations (Brezinski et al., 2022), where it often delivers substantial acceleration in computational tasks.

Applying the Anderson mixing method to the Hamiltonian inference problem, specifically to the Quantum Iterative Scaling (QIS) algorithm, is straightforward due to the inherent nature of QIS as a fixed-point iterative update algorithm. The Anderson-accelerated QIS (AM-QIS) algorithm combines both the QIS iterative step and simple classical processing, so it has exactly the same requirement as the standard QIS algorithm for the oracle access to the Gibbs state or the average values. Since the fixed-point map $g(x)$ in QIS iteration is a contraction, we can set the mixing parameter $\beta_t \equiv 1$ defined in Appendix D and the convergence of AM-QIS follows from the results in Toth & Kelley (2015). We also use the Barzilai-Borwein (BB) method (Barzilai & Borwein, 1988) for choosing the mixing parameter which turns out to be effective and provides further accelerations.

The BFGS method and the limited memory variant L-BFGS are the most influential among many quasi-Newton methods. They are the recommended choice for learning graphical models in the classical machine learning literature Malouf (2002). The BFGS method works with an unconstrained optimization problem $\min_{x \in \mathbb{R}^n} f(x)$. The update in the BFGS algorithm has the form $x_{k+1} = x_k - \eta_k H_k \nabla f(x_k)$, where $\eta_k$ is the step size which can usually be found by line search, and $H_k$ is a matrix that is updated iteratively during the execution of the algorithm. We consider both a fixed choice or the BB method for the initial approximation of the inverse Hessian $H_0$ for a fair comparison with AM. The application of BFGS methods to our problem is also straightforward as the dual problem is an unconstrained optimization problem.

AM and BFGS have different application scenarios. AM is an acceleration method for solving fixed-point problems and the approximation $G_t$ of the inverse Jacobian matrix is generally not symmetric. In contrast, BFGS is an optimization method and constructs a symmetric approximation $H_t$ for the inverse Hessian matrix. Since an optimization problem can usually be recast as a fixed-point problem, AM also applies to solving optimization problems. However, BFGS may be more efficient in some cases due to the maintained symmetry structure compared with AM.

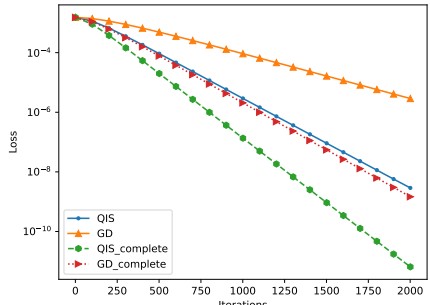

Figure 2: Comparison of QIS and GD algorithms. The loss in measured by the error in the objective function of the maximum entropy problem.

## 6 EXPERIMENTS

We conducted numerical simulations to assess the comparative efficiency of four approaches: the standard QIS, the standard GD algorithm, AM-QIS, and the L-BFGS-GD algorithm applied to the dual problem. In the experimental setup, we adopted a method involving generating random Gibbs states for random local Hamiltonians, represented as $H = \sum_j \lambda_j H_j$. Here, the local terms, denoted as $H_j$, consist of tensor products of local Pauli operators, and the $\lambda_j$ parameters are the values to be learned. These Hamiltonians were then utilized to create Gibbs states $\xi_H = \frac{1}{Z} e^{-\beta H}$. We feed the Gibbs states and their corresponding local average values $\alpha_j = \langle H_j \otimes \mathbb{1}, \xi_H \rangle$ to the algorithms. In this way, we know the

ground truth about the values of $\lambda_j$'s and the objective value of the optimization programs in Fig. 1, and we choose to evaluate the algorithms' performance by the error compared with the true objective value.

The results are summarized in Figs. 2 and 3. In Fig. 2, we compare the performance of QIS and GD algorithms. We can see that QIS algorithm is more efficient than GD algorithm regardless of whether we ensure the completeness $\sum F_j = \mathbb{1}$ or not. In Fig. 3, we compare the performance of AM-QIS and L-BFGS-GD, both with and without the Barzilai-Borwein method. We can see that AM-QIS and L-BFGS-GD are comparable in general. The standard QIS algorithm typically required approximately 1500 iterations to achieve an error level of $10^{-6} \sim 10^{-8}$ (measured using the objective function of the maximum entropy problem). In contrast, the AM-QIS and L-BFGS algorithms achieved the same accuracy with only about 8 (or 20) iterations with (or without) BB, showcasing a remarkable speedup of two orders of magnitude. The efficiency of the AM-QIS algorithm is stable and does not change much when the Hamiltonian is normalized and completed, while the efficiency of L-BFGS-GD algorithm (in Fig. 3a) is observed to be sensitive in this regard.

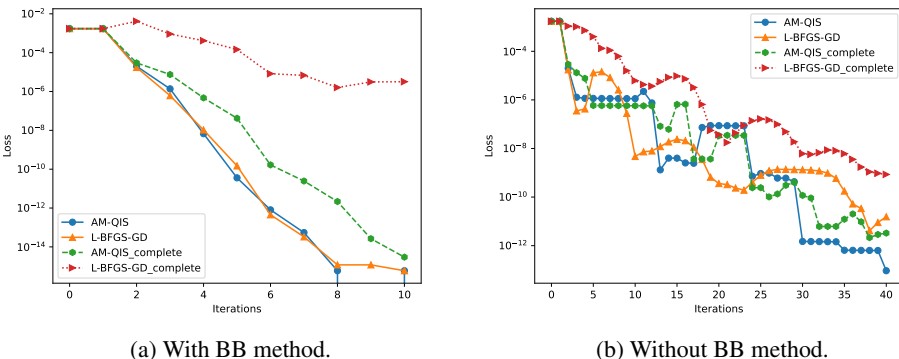

(a) With BB method.   (b) Without BB method.

Figure 3: Comparisons of the AM-QIS and L-BFGS-GD algorithms. The Fig. 3a on the left uses the Barzilai-Borwein method to choose the step size and Fig. 3b on the right uses fixed step size. The dotted (red) and dashed (green) lines represent the performance of the algorithms when the input Hamiltonian terms are complete satisfying $\sum_j F_j = \mathbb{1}$.

## 7 DISCUSSIONS

In this study, we considered adaptive learning algorithms for the Hamiltonian inference problem. We examined the convergence of the quantum iterative scaling algorithm (QIS) and the gradient descent (GD) algorithm for the dual problem. Furthermore, two quasi-Newton methods AM-QIS and F-BFGS-GD are proposed.

The QIS algorithm iteratively updates the Hamiltonian parameters adaptively by comparing $\left\langle H_j \otimes \mathbb{1}, \xi_{H(\lambda)} \right\rangle$ and the target value $\alpha_j$. Therefore it requires an oracle to be able to prepare $\xi_{H(\lambda)}$ for the trial parameter $\lambda$. The use of Gibbs state oracle is generally a computationally demanding assumption, but if the physical system has exponentially decaying correlation and satisfy certain Markov property, the preparation of the Gibbs state or its local observations could be efficient (Brandão & Kastoryano, 2019; Kuwahara et al., 2020). Furthermore, the issue may be solved or mitigated by combining quantum belief propagation algorithms proposed in Hastings (2007); Leifer & Poulin (2008); Poulin & Hastings (2011) which present possible ways of computing the value $\left\langle H_j \otimes \mathbb{1}, \xi_{H(\lambda)} \right\rangle$ approximately without generating the full Gibbs state, thereby removing the use of the adaptive Gibbs oracle. We leave the exploration of this possibility as future work.

In the proof of the upper bound of the Hessian of the log-partition function, we developed a modified quantum belief propagation technique, which may be of independent interest. It is an interesting problem to find more applications of this new tool.

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

## A    PRIME AND DUAL DIVERGENCE MINIMIZATION PROBLEMS

The primal and dual formulation of the maximum entropy program in Fig. 1 is a special case of the following duality result between two minimization problems of the Kullback-Leibler divergence given in Fig. 4. The duality theorem of Ji (2022), or Jaynes' principle, states that the following two problems have the same minimizer which is the unique intersection point of the linear family $\mathscr{L}(\rho_0)$ and exponential family $\mathrm{cl}\,(\mathscr{E}(\sigma_0))$ defined in Section 3. When $\sigma_0$ is the maximally mixed state $\mathbb{1}/d$, the linear family minimization is the maximum entropy problem and the exponential family minimization is a dual program in Fig. 1.

| Linear family minimization | Exponential family minimization |
|---|---|
| minimize:   $D(X, \sigma_0)$ | minimize:   $D(\rho_0, Y)$ |
| subject to:   $X \in \mathscr{L}(\rho_0)$. | subject to:   $Y \in \mathscr{E}(\sigma_0)$. |

Figure 4: Two optimization problems of the Kullback-Leibler divergence that are dual to each other.

## B    PROOFS FOR CONVERGENCE RATE

This section proves the explicit formulas for the Jacobian matrix of the iterations in QIS and GD algorithms.

*Proof of Theorem 4.1.* We first prove two identities about the partial derivative of the function $\ell$ and its natural logarithm.

$$\frac{\partial}{\partial \lambda_{j'}} \ell(\lambda \cdot F) = \langle F_{j'}, \exp(\lambda \cdot F) \rangle, \tag{2}$$

$$\frac{\partial}{\partial \lambda_{j'}} \ln \ell(\lambda \cdot F) = \langle F_{j'}, \xi \rangle, \tag{3}$$

for $\xi$ defined as

$$\xi = \frac{\exp(\lambda \cdot F)}{\mathrm{tr}\,\exp(\lambda \cdot F)}. \tag{4}$$

In fact, we have

$$\begin{aligned}
\frac{\partial}{\partial \lambda_{j'}} \ell(\lambda \cdot F) &= \frac{\partial}{\partial \lambda_{j'}}\,\mathrm{tr}\,\exp(\lambda \cdot F)\\
&= \mathrm{tr}\,\sum_{k=0}^{\infty} \frac{\partial}{\partial \lambda_{j'}} \frac{(\lambda \cdot F)^k}{k!}\\
&= \mathrm{tr}\,\sum_{k=1}^{\infty} \sum_{j=0}^{k-1} \frac{(\lambda \cdot F)^{k-j-1} F_{j'} (\lambda \cdot F)^j}{k!}\\
&= \left\langle F_{j'}, \sum_{k=1}^{\infty} \frac{(\lambda \cdot F)^{k-1}}{(k-1)!} \right\rangle\\
&= \langle F_{j'}, \exp(\lambda \cdot F) \rangle,
\end{aligned}$$

where the second line follows from the Taylor expansion of the matrix expansion and the fourth line is by the cyclic property of trace. This proves Eq. (2). Similarly, we have

$$\begin{aligned}
\frac{\partial}{\partial \lambda_{j'}} \ln \ell(\lambda \cdot F) &= \frac{1}{\ell(\lambda \cdot F)} \frac{\partial}{\partial \lambda_{j'}} \ell(\lambda \cdot F)\\
&= \frac{1}{\mathrm{tr}\,\exp(\lambda \cdot F)} \langle F_{j'}, \exp(\lambda \cdot F) \rangle\\
&= \langle F_{j'}, \xi \rangle,
\end{aligned}$$

where the second line follows from Eq. (2). This completes the proof of Eq. (3).

Now, for $j = 1, 2, \ldots, k$, the update in the algorithm is

$$\delta_j = \ln \langle F_j, \rho_0 \rangle - \ln \left\langle F_j, Y^{(t)} / \operatorname{tr} Y^{(t)} \right\rangle.$$

Hence, for $j, j' \in \{1, 2, \ldots, k\}$ and $\xi$ defined in Eq. (4),

$$\begin{aligned}
\frac{\partial \delta_j}{\partial \lambda_{j'}} &= -\frac{1}{\langle F_j, \xi \rangle} \frac{\partial \langle F_j, \xi \rangle}{\partial \lambda_{j'}} \\
&= -\frac{1}{\langle F_j, \xi \rangle} \frac{\partial}{\partial \lambda_{j'}} \frac{\partial}{\partial \lambda_j} \ln \ell(\lambda \cdot F) \\
&= -P_{j,j}^{-1} L_{j,j'}.
\end{aligned}$$

For the second line, we used Eq. (3). Equivalently, the Jacobian matrix

$$\left( \frac{\partial \delta_j}{\partial \lambda_{j'}} \right)_{j,j'} = -P^{-1} L,$$

for matrices $P$ and $L$ defined in the statement of the theorem. The Jacobian $J_{\mathrm{QIS}}$ of the QIS iteration can be written as

$$J_{\mathrm{QIS}} = \mathbb{1} + \left( \frac{\partial \delta_j}{\partial \lambda_{j'}} \right)_{j,j'} = \mathbb{1} - P^{-1} L.$$

This completes the proof the theorem. $\qquad\square$

*Proof of Theorem 4.2.* In an iteration of the algorithm, we have

$$\delta_j = \eta \langle F_j, \rho_0 \rangle - \eta \left\langle F_j, Y^{(t)} / \operatorname{tr} Y^{(t)} \right\rangle.$$

The Jacobian $J_{\mathrm{GD}}$ of each iteration has $(j, j')$ entry

$$\begin{aligned}
& \mathbb{1} - \eta \frac{\partial}{\partial \lambda_{j'}} \langle F_j, \xi \rangle \\
=\, & \mathbb{1} - \eta \frac{\partial}{\partial \lambda_{j'}} \frac{\partial}{\partial \lambda_j} \ln \operatorname{tr} \exp(\lambda \cdot F) \\
=\, & \mathbb{1} - \eta L.
\end{aligned}$$

This completes the proof. $\qquad\square$

We recall a theorem of Ostrowski which we will use to prove the convergence rate by bounding the spectral radius of a Jacobian matrix.

**Theorem B.1** (Ostrowski's theorem (Ostrowski, 1966, Chapter 22))**.** *Assume function $f$ is differentiable at the neighborhood of a fixed point $\zeta$. For an iterative algorithm $\zeta_{t+1} = f(\zeta_t)$. A sufficient condition for $\zeta$ to be a point of attraction is the spectral radius $r(J_f) < 1$. Moreover, if $\zeta$ is an attraction point, the geometric convergence rate of the iterative algorithm is given by*

$$\limsup_{t \to \infty} \frac{\|\zeta_{t+1} - \zeta\|}{\|\zeta_t - \zeta\|} = r(J_f).$$

## C  BOUNDS ON THE HESSIAN MATRIX

In this section, we prove the upper bound on the Hessian of the log-partition function.

The proof uses a modified quantum belief propagation. The idea of quantum belief propagation was studied in Hastings (2007) and we give a version of it in the following lemma. It specifies how the matrix exponential function changes with perturbations of the matrix.

**Lemma C.1** (Quantum Belief Propagation (Hastings, 2007)). *Suppose $f_\beta(t)$ is the function whose Fourier transform is*

$$\tilde{f}_\beta(\omega) = \frac{\tanh(\beta\omega/2)}{\beta\omega/2}$$

*and $H(s) = H + sV$ for $s \in [0, 1]$. Define the quantum belief propagation operator*

$$\Phi_{H(s)}(V) = \int_{-\infty}^{\infty} dt \, f_\beta(t) \, e^{-iH(s)t} V e^{iH(s)t}.$$

*Then*

$$\frac{d}{ds} \exp(\beta H(s)) = \frac{\beta}{2} \Big\{ \exp(\beta H(s)), \Phi_{H(s)}(V) \Big\}.$$

Here, we introduce a modified version of it to prove that if the perturbation is positive semidefinite, then so is the derivative of the matrix exponential function. That is, the modified quantum belief propagation expresses the derivative $\frac{d}{ds} \exp(\beta H(s))$ so that its positivity is obvious for positive $V$.

The proof uses the Bochner's theorem and we give a simple version of it which suffices for our purpose.

**Lemma C.2** (Bochner's Theorem). *A continuous function $f(x)$ on the real line with $f(0) = 1$ is positive-definite if and only if its Fourier transform is a probability measure on $\mathbb{R}$.*

**Lemma C.3.** *Suppose $g_\beta$ is a function whose Fourier transform is*

$$\tilde{g}_\beta(\omega) = \frac{e^{\beta\omega/2} - e^{-\beta\omega/2}}{\beta\omega}$$

*and $H(s) = H + sV$ for $s \in [0, 1]$. Define the modified quantum belief propagation operator*

$$\Psi_{H(s)}(V) = \int_{-\infty}^{\infty} dt \, g_\beta(t) \, e^{-iH(s)t} V e^{iH(s)t}.$$

*Then*

$$\frac{d}{ds} \exp(\beta H(s)) = \beta \, \exp\Big(\frac{\beta H(s)}{2}\Big) \, \Psi_{H(s)}(V) \, \exp\Big(\frac{\beta H(s)}{2}\Big). \tag{5}$$

*Furthermore, $g_\beta(t)$ is a probability density function over the real line and $\Psi_{H(s)}$ is a completely positive trace-preserving map.*

*Proof of Lemma C.3.* Consider the spectrum decomposition of $H(s)$ as $H(s) = \sum_j \lambda_j |\psi_j\rangle\langle\psi_j|$.

Using Duhamel's formula, we have

$$\frac{d}{ds} \exp(\beta H(s)) = \int_0^1 dt \, e^{t\beta H(s)} \Big(\frac{d}{ds}\beta H(s)\Big) e^{(1-t)\beta H(s)}$$

$$= \beta \int_0^1 dt \, e^{t\beta H(s)} V e^{(1-t)\beta H(s)}.$$

Hence, the $(j, j')$-th entry of $\frac{d}{ds} \exp(\beta H(s))$ in the basis of $\{|\psi_j\rangle\}$ is

$$\langle\psi_j| \frac{d}{ds} \exp(\beta H(s)) |\psi_{j'}\rangle = \beta V_{j,j'} \int_0^1 dt \, e^{t\beta\lambda_j + (1-t)\beta\lambda_{j'}}$$

$$= \begin{cases} \dfrac{e^{\beta\lambda_j} - e^{\beta\lambda_{j'}}}{\lambda_j - \lambda_{j'}} V_{j,j'} & \text{if } \lambda_j \neq \lambda_{j'} \\ \beta e^{\beta\lambda_j} V_{j,j'} & \text{o.w.} \end{cases} \tag{6}$$

where $V_{j,j'} = \langle\psi_j|V|\psi_{j'}\rangle$.

Now we simplify the right-hand side of Eq. (5). By the definition of $\Psi_{H(s)}(V)$, the $(j, j')$-th entry of $\Psi_{H(s)}(V)$ in the basis $\{|\psi_j\rangle\}$ is

$$\int_{-\infty}^{\infty} dt \, g_\beta(t) \, e^{-i\lambda_j t} V_{j,j'} e^{i\lambda_{j'} t} = \tilde{g}_\beta(\lambda_j - \lambda_{j'}) \, V_{j,j'}.$$

Hence, the $(j, j')$-th matrix entry of right-hand side in the basis $\{|\psi_j\rangle\}$ can be written as

$$
\beta\, e^{\beta(\lambda_j+\lambda_{j'})/2}\, \tilde{g}_\beta(\lambda_j - \lambda_{j'})\, V_{j,j'} =
\begin{cases}
\dfrac{e^{\beta\lambda_j} - e^{\beta\lambda_{j'}}}{\lambda_j - \lambda_{j'}} V_{j,j'} & \text{if } \lambda \neq \lambda_{j'} \\
\beta e^{\beta\lambda_j} V_{j,j'} & \text{o.w.}
\end{cases}
$$

which is the same as the $(j, j')$-th entry of the left-hand side by Eq. (6). This completes the proof of Eq. (5). The fact that $g_\beta(t)$ is a probability density function and the CPTP property of $\Psi_{H(s)}$ follow from the Bochner's theorem applied to $g_\beta$ and $\tilde{g}_\beta$ and the fact that $\tilde{g}_\beta(0) = 1$. $\qquad\square$

To prove the bound in Theorem 4.3, we need two related results proved in Theorem C.1 and Lemma C.4 which we now prove.

Define matrices

$$
\Delta = \sum_j \frac{\partial \ell(\lambda \cdot F)}{\partial \lambda_j} |j\rangle\langle j|,
$$

$$
\Lambda = \sum_{j,j'} \frac{\partial^2 \ell(\lambda \cdot F)}{\partial \lambda_j\, \partial \lambda_{j'}} |j\rangle\langle j'|.
$$

$\Delta$ is a diagonal matrix and $\Lambda$ is the Hessian of the partition function $Z = \ell(\lambda \cdot F) = \operatorname{tr} \exp(\lambda \cdot F)$.

**Theorem C.1.** *For $\Lambda$ and $\Delta$ defined above, we have $\Lambda \preceq \Delta$.*

*Proof of Theorem C.1.* Choose $H = \ln Y_0 + \lambda \cdot F$, $\beta = 1$, $s = 0$, and $V = F_{j'}$ in Lemma C.3, we have

$$
\begin{aligned}
\Lambda_{j,j'} &= \left\langle F_j, \frac{\partial}{\partial \lambda_{j'}} \exp(H) \right\rangle \\
&= \left\langle F_j, e^{H/2}\, \Psi_H(F_{j'})\, e^{H/2} \right\rangle \geq 0.
\end{aligned}
$$

That is, all entries of matrix $\Lambda$ (in the basis $(|\psi_j\rangle)$) are non-negative.

Next, we prove that $\Delta - \Lambda$ is a diagonally dominant matrix. For all $j'$, the sum of the $j'$-th column is

$$
\begin{aligned}
\Delta_{j',j'} - \sum_j \Lambda_{j,j'} &= \left\langle F_{j'}, e^H \right\rangle - \left\langle \sum_j F_j, e^{H/2}\Psi_H(F_{j'})e^{H/2} \right\rangle \\
&\geq \left\langle F_{j'}, e^H \right\rangle - \left\langle \Psi_H(F_{j'}), e^H \right\rangle \\
&= \left\langle F_{j'}, e^H \right\rangle - \left\langle \int_{-\infty}^{\infty} \mathrm{d}t\, g_1(t)e^{-iHt}F_{j'}e^{iHt}, e^H \right\rangle \\
&= \left\langle F_{j'}, e^H \right\rangle - \int_{-\infty}^{\infty} \mathrm{d}t\, g_1(t)\, \operatorname{Tr}\!\left(e^{-iHt}F_{j'}e^{iHt}e^H\right) \\
&= \left\langle F_{j'}, e^H \right\rangle - \int_{-\infty}^{\infty} \mathrm{d}t\, g_1(t)\, \operatorname{Tr}\!\left(F_{j'}e^H\right) \\
&= \left\langle F_{j'}, e^H \right\rangle - \left\langle F_{j'}, e^H \right\rangle = 0.
\end{aligned}
$$

In the above, the inequality follows from $\sum_j F_j \preceq \mathbb{1}$, the positivity of $e^{H/2}\Psi_H(F_j)e^{H/2}$ and the cyclic property of the trace. The fifth line uses the commutativity of $e^{iHt}$ and $e^H$ and the cyclic property of the trace. The last line follows from the fact that $g_1(t)$ is the probability density function by the Bochner's theorem.

The claim in the theorem now follows by the well-known matrix theory result that diagonally dominant matrices are positive semidefinite. $\qquad\square$

**Lemma C.4.** *For matrices $\Delta, \Lambda$ defined above and $Q = \sum_{j,j'} \langle F_j\rangle \langle F_{j'}\rangle |j\rangle\langle j'|$, we have the following identity*

$$
\Lambda = Z(L + Q).
$$

*Proof.* By definition, we have

$$
\begin{aligned}
L_{j,j'} &= \frac{\partial^2}{\partial \lambda_j \, \partial \lambda_{j'}} \ln Z \\
&= \frac{1}{Z^2} \Big( Z \frac{\partial^2 Z}{\partial \lambda_j \, \partial \lambda_{j'}} - \frac{\partial Z}{\partial \lambda_j} \frac{\partial Z}{\partial \lambda_{j'}} \Big) \\
&= \frac{1}{Z} \Lambda_{j,j'} - \langle F_j \rangle \langle F_{j'} \rangle,
\end{aligned}
$$

or equivalently $\Lambda = Z(L + Q)$ in the matrix form. $\qquad\square$

We are now ready to prove the main result stated in Theorem 4.3.

*Proof of Theorem 4.3.* By Lemma C.4, we have

$$
L = \frac{\Lambda}{Z} - Q.
$$

Hence, Theorem C.1 implies that

$$
L = \frac{\Lambda}{Z} - Q \preceq \frac{\Delta}{Z} - Q = P - Q.
$$

As $Q$ is outer product of vector $\sum_j \langle F_j \rangle |j\rangle$, it is a rank-1 and positive semidefinite matrix. Therefore, we have

$$
L \preceq P - Q \preceq P
$$

which completes the proof. $\qquad\square$

Finally, we will also need a lower bound on $L$, for which we recall a result about the strong convexity of the log-partition function from Anshu et al. (2021).

**Theorem C.2** (Theorem 6 of Anshu et al. (2021)). *Let $H(\mu) = \sum_{j=1}^m \mu_j H_j$ be an $\ell$-local Hamiltonian over a finite dimension lattice. For a given inverse temperature $\beta$, there are constants $c, c' > 3$ depending on the geometric property of the lattice such that*

$$
\nabla_\mu^2 \ln \mathrm{tr}\big(e^{-\beta H(\mu)}\big) \succeq \frac{e^{-O(\beta^c)} \beta^{c'}}{m} \mathbb{1}.
$$

## D   DISCUSSIONS ON QUASI-NEWTON METHODS

In this section, we give some details of the Anderson mixing method and L-BFGS method.

The algorithm interpolates history information in order to speed up a fixed-point iteration. More concretely, suppose $g : \mathbb{R}^d \to \mathbb{R}^d$ is a contraction and we are interested in finding the fix-point $x = g(x)$. The standard fix-point iterative algorithm is to compute $x_{t+1} = g(x_t)$ for $t = 0, 1, 2, \ldots$, until a stopping criteria is met. Anderson mixing accelerates the iteration by using the history information of the previous iterative steps. A relatively small history size $m \geq 0$ is chosen and we define $m_t = \min\{m, t\}$. In our numerical implementation, we use $m = 10$. Define the residual $r_t = g(x_t) - x_t$ and two matrices $X_t, R_t \in \mathbb{R}^{d \times m}$ storing the historical information

$$
\begin{aligned}
X_t &= \big(\Delta x_{t-m_t}, \Delta x_{t-m_t+1}, \ldots, \Delta x_{t-1}\big), \\
R_t &= \big(\Delta r_{t-m_t}, \Delta r_{t-m_t+1}, \ldots, \Delta r_{t-1}\big).
\end{aligned}
$$

Then, the Anderson accelerated iteration can be written succinctly as $x_{t+1} = x_t + G_t r_t$ where

$$
G_t = \beta_t I - (X_t + \beta_t R_t)\big(R_t^T R_t\big)^{-1} R_t^T.
$$

Here, $\beta_t$ is the mixing parameter.

It is pointed out in Fang & Saad (2009) that $G_t$ approximates the inverse of the Jacobian of $g$ and Anderson mixing method can be thought of as a quasi-Newton method satisfying multi-secant equations. We note that there is a matrix inverse in the above formula for $G_t$ which can be implemented

using Moore-Penrose pseudo-inverse. For stability and efficiency concerns, we found that the AM algorithms have the best performance in our numerical simulations when using a relative condition number of $1e-7$ in the pseudo-inverse, a cutoff threshold that sets small singular values of the matrix to zero. This is easily implemented by setting the rcond parameter of the pinv function in numpy.linalg package for Python implementations.

The BFGS method is one of the most popular quasi-Newton methods that can be applied to unconstrained optimization problems $\min_{x\in\mathbb{R}^n} f(x)$. It is also known as the variable metric algorithm as first proposed by Davidon (Davidon, 1991; Yuan, 2015). The algorithm maintains the approximate Hessian $H_{k+1}$ of the optimization problem. The update rule of the algorithm is

$$x_{t+1} = x_t - \eta_t H_t \nabla f(x_t),$$

and the update rule of $H_t$ is

$$H_{t+1} = \left(\mathbb{1} - \frac{s_t y_t^T}{y_t^T s_t}\right) H_t \left(\mathbb{1} - \frac{y_t s_t^T}{y_t^T s_t}\right) + \frac{s_t s_t^T}{y_t^T s_t},$$

where $s_t = x_{t+1} - x_t$, $y_t = \nabla f(x_{t+1}) - \nabla f(x_t)$ and $H_0$ is a predefined initial approximation of the inverse Hessian matrix. We refer readers to Nocedal & Wright (2006) for a discussion on how the BFGS update is derived. To optimize the memory usage, the limited memory version of BFGS called L-BFGS (Liu & Nocedal, 1989) is used. Since using line search for choosing $\eta_t$ can incur additional function evaluations in each iteration, we use $\eta_t \equiv 1$ and only tune $H_0$ in the numerical simulation.

Both the AM and the BFGS methods employed our numerical simulation can be further strengthened by using a heuristics invented by Barzilai and Borwein (Barzilai & Borwein, 1988) to choose the $\beta_t$ in AM and $H_0$ in BFGS. In AM, we set

$$\beta_t = -\frac{\Delta r_{t-1}^T \Delta x_{t-1}}{\Delta r_{t-1}^T \Delta r_{t-1}},$$

which solves $\min_\beta \|\Delta x_{t-1} + \beta \Delta r_{t-1}\|_2$. In BFGS, we set

$$H_0 = \frac{y_{t-1}^T s_{t-1}}{y_{t-1}^T y_{t-1}} \mathbb{1}$$

to be the initialization of the approximate inverse Hessian in the $t$-th iteration (Nocedal & Wright, 2006, Pages 143 and 178).

