# OpenReview forum: "Adaptive Learning of Quantum Hamiltonians"
_ICLR.cc/2024/Conference — Submitted to ICLR 2024_

### Official Review · Reviewer_yC5R · 2023-10-16

**Soundness:** 3 good
**Presentation:** 3 good
**Contribution:** 2 fair
**Rating:** 3
**Confidence:** 5

**Summary:**

This work builds on recent breakthroughs in learning of quantum gibbs states from measurement data. The authors apply new optimization methods to solve the max-entropy problem associated with this learning task. They also derive rigorous complexity bounds for these optimization methods. The main technical contribution is a modified version of the Quantum Belief Propagation technique that is used to upper bound the eigenvalues of the Hessian.

**Strengths:**

1.  Gives rigorous guarantees for the computational complexity of learning quantum Hamiltonains from solving the max entropy problem.

2. The new Quantum BP method in this work could be a useful alternative to the method introduced by Hastings. If the authors can find some use cases where this method gives better bounds then this would be a good addition to the quantum information literature

**Weaknesses:**

1. Oracle assumed is too strong and impractical. The oracle solves a very hard computational problem even in the classical setting, especially in the low temperature regime. The setting in this work is the quantum analogue of learning classical graphical models given an oracle that can always return the sufficient statistics given the energy function of a model.

The main technical challenge that needs to be addressed in this field is the fact that there are no quantum versions of the pseudo-likelihood type methods that are used to learn classical Gibbs states. The existence of these methods in the quantum regime would bypass the need for these types of oracles. A very recent work (https://arxiv.org/abs/2310.02243) has shown that the computational complexity of solving this problem is polynomial in the size of the system, without assuming any strong oracles. This new method has much worse sample complexity, so a practical algorithm is still out of reach.

In general oracles are useful to establish relative complexity of two tasks, inference and learning in this case. But for learning Gibbs states, from classical results, we already know that inference is much harder than learning. So the bounds established in this work are only marginally interesting.

2. It is mostly using established techniques in optimization to solve a standard problem of matching sufficient statistics (essentially Maximum Likelihood).  Novelty of this work mainly comes from the techniques used to establish the bounds on Hessian.  The methods used to solve the problem it self are not new.

3. Overall the method is not very efficient. Theory gives an exponential worst case run-time (which is absorbed into the oracle) and the field of quantum computing is too nascent to provide any high quality data  to establish any practical claims regarding the algorithm.

Overall I don't think this work has made any good breakthroughs in solving this problem and is not a good fit for ICLR. The new  Quantum BP method is interesting and might be a good research direction.

**Questions:**

1. Are there useful cases of Gibbs state learning where some algorithm can implement the oracle used in this work without directly preparing the Gibbs states?

2. How does the complexity of this work compare to the high-temperature learning results of Haah et al. (https://arxiv.org/abs/2108.04842)?
The oracle used in the paper under review can possibly be implemented efficiently in this regime. But then the total complexity of the method proposed here must be compared with that of Haah et al.

---

> ### Author Response · Authors · 2023-11-19
> **Reply to reviewer vC5R**
>
> We thank the reviewer for a very informative report.
>
> The reviewer's primary concern is the use of the adaptive oracle access model. Especially, the reviewer thought that the oracles assumed in this paper (referred to as adaptive oracles in the reply) were stronger than those in the Hamiltonian learning from Gibbs states (referred to as Gibbs oracles).
> While adding adaptivity to the oracle does strengthen the requirements, we do not believe that the adaptive oracle is significantly different from the Gibbs oracle for the following reason. Both setups necessitate access to copies of Gibbs states or their local density matrices associated with a specific Hamiltonian. In the Hamiltonian learning from Gibbs states problem, the oracle is implicit in the problem, assuming that nature or a specific algorithm can prepare and provide such states for the true Hamiltonian. In the adaptive oracle setup, the Hamiltonian is adaptively chosen, but it shares the same locality as the true Hamiltonian. Thus, assuming a natural process or algorithm exists for preparing the true Hamiltonian, the same process could be employed for the adaptively chosen Hamiltonian. In this sense, it would be inaccurate to claim that Hamiltonian learning algorithms do not require strong oracles and that the adaptive oracle is significantly stronger than Gibbs oracles. We do agree with the reviewer that it is an important issue that needs to be addressed, and exploring under what conditions we can approximate the local density matrices of the Gibbs state is a fascinating question. This may, indeed, limit the algorithm to certain special cases where such an approximation is feasible.
>
> The reviewer also said, "... from classical results, we know that inference is much harder than learning". This probably refers to the result of Montanari 2015. The claim is correct for "inference from sufficient statistics" and may not apply to the more general setting. The all-temperature learning algorithm (arXiv:2310.02243) the reviewer mentioned can be considered an *inference* algorithm where the data used is 2^\beta-local density matrices of the Gibbs state. In other words, it is possible to consider an inference algorithm where the local regions are larger than the locality of the Hamiltonian.
>
> The reviewer asks for a comparison of the complexity of our algorithms with high-temperature learning results of arXiv:2108.04842. We will add a comparison in future revisions. However, we would like to emphasize that the focus of the paper is not to improve theoretical lower bounds at the expense of algorithm simplicity. Our paper aims to consider simpler algorithms grounded in classical theory and demonstrate how optimization heuristics can further enhance their practicality.

---

> ### Comment · Reviewer_yC5R · 2023-11-20
>
> Thanks to the authors for responding to the review.
> I still believe that the adaptive oracle is much stronger than the Gibbs oracle, as it can prepare states from a phase that is different from the state that the algorithm is trying to learn. For instance, consider a classical version of this problem. Having a system that can prepare the high temperature  state of an Ising type model with frustration is much simpler that constructing an oracle that can return expectation values from the same model at a lower temperature (for instance in a spin glass phase). Even natural processes can get stuck in metastable states at lower temperatures in such models. So it cannot be assumed that natural process that prepares a high-temperature state can be used to generate states of the same Hamiltonian at a lower temperature. The main weakness I see in the paper is that it is solving a strictly easier problem than Anshu et. al (2021) and I do not see any good reason to consider the weaker version of the problem.

---

> > ### Author Response · Authors · 2023-11-23
> > **Reply to reviewer**
> >
> > We appreciate the reviewer's feedback.
> >
> > Regarding the phase transition argument, we are convinced that adaptive oracles may be stronger in general. Thanks! However, in our algorithm, we initialize the Hamiltonian parameters with all zeros, corresponding to the infinite temperature case. The Hamiltonian then undergoes iterative updates towards the true Hamiltonian. Based on this process, we expect that the path of the Hamiltonian will remain in the same phase for most cases.
> >
> > We do not see why the reviewer claimed that the problem we solve is strictly easier. Please see the discussion in the paper about these two problems for reasons that we consider them to be incomparable. Probably the reviewer was referring to the adaptive oracle again. We believe the study under this model is valuable for the following reasons.
> >
> > First, the classical analogs of the algorithms we have used are state-of-the-art for the corresponding classical counterpart problems. The criticism regarding the adaptive oracle also applies to the classical case. In classical models, we also observe phase transition phenomena when the temperature changes, yet these classical algorithms have been proven to be valuable in practice. We expect a similar situation in the quantum case.
> >
> > Second, our algorithm is much simpler compared to other existing quantum Hamiltonian learning algorithms. The simplicity of our algorithm also ensures that the quasi-Newton accelerations discussed in the paper perform well. This work is probably the first to consider quasi-Newton speedups in the Hamiltonian learning literature. We believe this is a must if any quantum Hamiltonian learning algorithm is ever used in practice because of its dramatic savings in computation time.
> >
> > Third, quantum generalizations are significantly harder to analyze, and it is not clear that classical algorithms can be naturally generalized to the quantum case when the Hamiltonians are non-commutative. Our work developed techniques and methods to resolve the difficulties partially.
> >
> > Anyway, we thank the reviewer for the helpful discussions. We will clarify the points in future revisions of the paper.

---

> > > ### Comment · Reviewer_yC5R · 2023-11-23
> > >
> > > Thank you for the detailed response

---

### Official Review · Reviewer_LkFc · 2023-10-29

**Soundness:** 3 good
**Presentation:** 3 good
**Contribution:** 2 fair
**Rating:** 5
**Confidence:** 3

**Summary:**

The paper considers a new problem of Hamiltonian inference. Here a learner would like to infer the coefficients of a local Hamiltonian. The oracle returns the expectation of any specific chosen product term of the Hamiltonian under the Gibbs state corresponding to a chosen Hamiltonian with similar geometry. The paper proposes two algorithms for this problem, namely the quantum iterative scaling and gradient descent, and proves their polynomial (in the number of the product terms of the Hamiltonian) convergence. Furthermore, the paper proposes two accelerations, based on quasi-Newton methods, to improve the convergence.

**Strengths:**

**significance:** The problem of Hamiltonian learning is fundamental in quantum learning.

**originality:** As far as I know, the Hamiltonian inference problem formulation of this paper is completely new. Moreover, the main technical tool is a new 'quantum belief propagation' lemma that can be important for other applications.

**quality:** The polynomial convergence of the proposed algorithm is good, as the number of product terms is itself polynomial in the number of qubits. These claims are supported by rigorous proofs. Finally, experiments clearly demonstrate the advantages of the proposed accelerations.

**clarity:**  The paper is written well overall. The ideas in the paper flow smoothly and are supported by good motivation. Additionally, the results presented in the paper are generally accompanied by thorough and informative comments or explanations.

**Weaknesses:**

The primary limitation of this paper is that the main algorithms are essentially specific instances of the quantum iterative scaling problem of Ji (2022) and gradient descent. While it's worth noting that their convergence analysis is not entirely straightforward, it relies on several pre-existing results, including Ostrowski's theorem and Anshu et al.'s (2021) lower bound on the Hessian $L$. Furthermore, the new 'quantum belief propagation' introduced in this paper ( cf. Lemma C.3) is similar to the original one, as can be seen by changing $\tanh\rightarrow \sinh$.

**Questions:**

Can you show a matching lower bound on adaptive algorithms for the Hamiltonian inference problem with the oracle you propose in the paper?

---

> ### Author Response · Authors · 2023-11-19
> **Reply to reviewer LkFc**
>
> We appreciate the reviewer's comments. While it is true that the algorithms we studied were proposed in recent works, the main contributions of our paper lie in (1) the analysis of the convergence rate of these algorithms in the quantum setting and (2) the exploration of quasi-Newton accelerations, specifically for the quantum case. Our motivation is to understand the quantum analogs of state-of-the-art classical algorithms and demonstrate their efficacy with provable convergence rates in the quantum domain. Additionally, we investigate the suitability of quasi-Newton accelerations for quantum scenarios.
>
> We acknowledge the reviewer's observation that our modified quantum belief propagation simply involves changing "tanh" to "sinh." However, the significance lies not in using a new function per se but rather in the fact that only the "sinh" version enables us to establish the desired upper bound for the Hessian of the log-partition function. It is natural to attempt to use the standard quantum belief propagation with "tanh" to prove this bound, but one would encounter difficulties establishing the derivative's positivity. While we did not attempt to establish a lower bound, we conjecture that similar bounds as in the Hamiltonian learning case would follow.
>
> Thank you for your valuable feedback, and we will ensure that these points are further clarified in the paper.

---

> > ### Comment · Reviewer_LkFc · 2023-11-22
> >
> > Thank you for your response.

---

### Official Review · Reviewer_VgoS · 2023-10-30

**Soundness:** 3 good
**Presentation:** 2 fair
**Contribution:** 2 fair
**Rating:** 6
**Confidence:** 3

**Summary:**

The paper studies quantum iterative scaling and gradient descent algorithms to learn the Hamiltonian of a quantum system by solving the maximum entropy problem or its dual.
They assume the setting where one has access to the expectation values of the terms of the Hamiltonian in the Gibbs state, as well as an oracle to prepare a Gibbs state adaptively to prescribed values of the Hamiltonian parameters.
The main result is to compute the converge rate of the algorithms as a function of the number of parameters. They also discuss acceleration of the algorithms by quasi Newton methods, that are supported by experiments showing the improvements.

**Strengths:**

- Well written
- Rigorous convergence analysis
- Novel bound on the Hessian of the free energy
- Practical consideration on acceleration by quasi-Newton methods

**Weaknesses:**

- Problem setting is not clearly motivated. While Hamiltonian learning is a central problem in quantum computing, it is not clear to me why one would prefer to look at the Hamiltonian inference problem where one has to prepare adaptively a Gibbs state and measure observables in it
- Preparing adaptively Gibbs states and measure observables in it is computationally hard - at least beyond some critical temperature, as discussed by the authors in the conclusions. See also the very recent work [https://arxiv.org/abs/2310.02243] for an efficient algorithm at all temperatures. It is then not clear whether this approach scales and is practical.
- I could not find details of the systems studied in the experiments.
- Novelty is limited as it is an extension of the framework of [Anshu et al] to bound the spectrum of the Hessian

Minor:
- Page 1: I think that there is no $\times 1$ in the definition of $\alpha_j$ since $H_j$ is already acting on the whole Hilbert space
- Line 4, Algorithm 1: I think k should be m

**Questions:**

- Can you add more details on the motivation and benefits of the Hamiltonian inference problem?
- What is the system size you used in the experiments?

---

> ### Author Response · Authors · 2023-11-19
> **Reply to reviewer VgoS**
>
> We appreciate the reviewer's thorough feedback. Here are our responses to the questions and weaknesses mentioned in the report.
>
> In terms of motivation, our goal is to explore the applicability of state-of-the-art classical algorithms to analogous problems in the quantum domain. We believe these classically inspired algorithms may offer practical advantages over existing quantum algorithms.
>
> We acknowledge the concern raised by the reviewer regarding the use of adaptive oracles, which was also questioned by another reviewer. While we agree that this is an area for improvement, we believe that the adaptive oracle assumption is only a slight strengthening of assuming the availability of Gibbs states for the true Hamiltonian. We argue that if there exists a natural process or algorithm capable of providing the Gibbs state for the true Hamiltonian (as required by the Hamiltonian learning problems), the same process should work for the adaptively chosen Hamiltonians in our algorithm, as these chosen Hamiltonians share the same locality as the true Hamiltonian.
>
> We are unsure what it refers to in your question about "systems studied" in the experiments. If the question pertains to the Hamiltonian systems considered in our numerical simulations, they are random 2-local Hamiltonians. If the reviewer was inquiring about the computing systems used for the experiments, we used a workstation with 32 CPUs (Intel(R) Xeon(R) Platinum 8369HB CPU @ 3.30GHz) and 64GB of memory.
>
> Regarding the novelty of our work, we indeed employ the work of Anshu et al. for the lower bound, while the upper bound of the spectrum serves as the main technical contribution of our paper.
>
> Regarding the minor comments:
>
> 1. We will clarify on page 1 that H_j refers to a local matrix. In local Hamiltonian literature, omitting the tensor I part for each H_j is customary when writing H = \sum_j H_j. We will ensure proper clarification in the next update.
>
> 2. Regarding the issue with Algorithm 1, we believe that using k is correct, as it represents the number of F_j operators.

---

> > ### Comment · Reviewer_VgoS · 2023-11-20
> >
> > Thank you for your response. I do not have further comments.

---

### Official Review · Reviewer_Zhky · 2023-11-01

**Soundness:** 3 good
**Presentation:** 3 good
**Contribution:** 2 fair
**Rating:** 5
**Confidence:** 3

**Summary:**

This paper studies the learning of quantum Hamiltonians using adaptive Gibbs state oracles. This paper's approach is based on quantum iterative scaling (QIS) and gradient descent (GD) with a more tailored analysis and closed-form formula for the
Jacobian.   The authors modified convergence rate analysis and studied heuristic algorithms for acceleration, including the quasi-Newton method. The claims have been supported by numerical experiments.

**Strengths:**

The main strength of the paper is in providing explicit Jacobian expressions for the adaptive algorithms used in Hamiltonian learning. Moreover, the convergence rate analysis derived in this paper is beneficial in understanding the Hamiltonian learning problem.

**Weaknesses:**

It seems that the paper's novelty is limited to some extent. Major results rely heavily on existing works such as Liang et al. (2004). Specifically, the main theorems in Section 4 are extensions of the analysis in the classical setting. It appears that the paper's results are restricted to the problem they study, and the contribution is only to provide an explicit formula for the Jacobian.

**Questions:**

Can you further explain the main novelty of the analysis compared to that of the classical settings?

---

> ### Author Response · Authors · 2023-11-19
> **Reply to reviewer Zhky**
>
> We would like to express our gratitude to the reviewer for their comments and questions. The generalization of classical results to the quantum case encounters a significant challenge due to the non-commutativity of matrices within the matrix exponentials present in the log-partition function. This non-commutativity prevents the derivative of the matrix exponential function from having a simple closed-form formula. The technical innovation of our paper lies in the application of a newly proposed quantum belief propagation method (Lemma C.3) to provide bounds on the eigenvalues of the Hessian of the log-partition function, even in cases where we do not possess a formula for the Hessian. In classical scenarios, the corresponding part of the proof would be relatively straightforward.

---

> > ### Comment · Reviewer_Zhky · 2023-11-20
> > **Thank you for your response**
> >
> > Thank you for your response. Unfortunately, I will keep my response.

---

### Meta-Review · Area_Chair_vjbw · 2023-12-09

**Metareview:**

This paper studies the learning of quantum Hamiltonians using adaptive learning. The reviewers are concerned with the technical advances of this work, but acknowledge the problem is interesting. Given ICLR is a technical conference, a reject is appropriate in this case.

**Justification For Why Not Higher Score:**

The reviewers are concerned with the technical advances of this work.

**Justification For Why Not Lower Score:**

NA

---

### Decision · Program_Chairs · 2024-01-16

Reject